# Freezing Effect and Bystander Effect: Overlaps and Differences

**Elena Siligato** [1], **Giada Iuele** [1], **Martina Barbera** [1], **Francesca Bruno** [1], **Guendalina Tordonato** [1], **Aurora Mautone** [1] **and Amelia Rizzo** [1,2,*]

1    Department of Clinical and Experimental Medicine, University of Messina, 98122 Messina, Italy; elenasiligato@libero.it (E.S.); giadaiuele@gmail.com (G.I.); martinabarbera17@gmail.com (M.B.); francescabruno.psi@outlook.it (F.B.); t.guenda@outlook.it (G.T.); aurora.mautone.25@gmail.com (A.M.)
2    Department of Cognitive Sciences, Psychology, Educational, and Cultural Studies, University of Messina, 98122 Messina, Italy
*    Correspondence: amrizzo@unime.it

**Abstract:** The present article provides a detailed comparison of two psychological phenomena, the freezing effect and the bystander effect, across their neurobiological, cognitive, emotional, and behavioral dimensions. This study focuses on identifying and analyzing the similarities and differences between these two responses to stressful and traumatic events. While the freezing effect is characterized by an involuntary neurobiological response to immediate threats, resulting in temporary immobilization or paralysis, the bystander effect describes a cognitive and social phenomenon where individuals refrain from offering help in emergencies when others are present. The study explores affective aspects, including emotional responses and trauma-related impacts associated with both phenomena. Through a comparative analysis, this research unveils important understandings regarding the distinctions among these responses, including their triggers, underlying mechanisms, and observable behaviors. It also highlights overlapping aspects, particularly in how both phenomena can lead to inaction in critical moments. This comparative study contributes to a deeper understanding of the complex interaction between the brain, individual cognition, and social dynamics in the face of danger and stress. The findings of this research have significant implications for understanding human behavior in emergencies, offering valuable perspectives that can be applied in the domains of psychology, training for emergency response, and trauma therapy.

**Keywords:** freezing effect; bystander effect; trauma

## 1. Introduction

The bystander effect and freezing phenomenon share similarities in that they result in inaction and an inability to intervene. However, they also have distinct differences. Although both are significant in social psychology, there has been a lack of systematic comparison between these two concepts. This article aims to fill this gap by comparing these mechanisms in detail. It will explore their definitions, emotional and behavioral responses, neurobiological reactions, and psychological outcomes, highlighting both their commonalities and unique aspects.

## 2. The Freezing Effect

Gray's hypothesis posits a neurobehavioral mechanism that governs human defensive behaviors in response to both innate and learned threat stimuli, with these responses phenomenologically correlating with the subjective experience of fear [1]. His theory aimed to deepen the relationship between the neurobiological systems regulating approach and avoidance behaviors. As known, Cannon [2] wrote about these two strategies of approaching threatening events/stimuli in emergency situations, which were later elaborated by Gray in the initial version of his theory as the FFS (fight–flight system), alongside two other response systems to threatening stimuli: the BAS (behavioral activation system) and the BIS

(behavioral inhibition system). The FFS was revised in 2000 by Gray and MacNaughton [3], becoming the FFFS (fight–flight–freeze system), thus incorporating the behavioral response of immobilization (freezing) in relation to aversive stimuli. This third option is characterized by an automatic alert action associated with fear, with postural stiffening and a decrease in cardiovascular and somatomotor activity [1,3]. Evolutionarily, indeed, prey that remains immobilized during a threatening situation has much less of a chance of being captured. This mechanism constitutes a key component of the reinforcement sensitivity theory (RST), which pertains to personality. The fight–flight–freeze system, operational upon the perception of threat, orchestrates adaptive responses to aversive stimuli, with fear being the subjective psychological state accompanying these responses. This system is delineated by a repertoire of defensive behaviors, which encompasses immobilization (freezing), rapid evasion (flight), and aggressive confrontation (fight) [4].

As a passive and defensive response to a stressful event, freezing is characterized by a reduction in body movements, bradycardia (a decrease in heart rate), and an increase in muscle tone [5]. The phenomenon of freezing is commonly linked with fear and is believed to enhance processes related to perception and attention, which help in identifying signals that dictate suitable actions [6,7]. Unlike extensive studies on freezing in animals and investigations into other human stress responses like fight or flight, research focusing on human freezing is quite limited. The concept of fight or flight as a human reaction to stress was established in the 1920s, but the idea of freezing as a third response only gained attention about half a century later and has not been thoroughly explored. In the animal kingdom, freezing in response to threats can be seen as an effective tactic, akin to feigning death in dangerous situations. In humans, however, freezing often translates to a paralysis of sorts, marked by an inability to communicate, respond, or engage in any actions for self-defense or preservation [8–12] (see Figure 1).

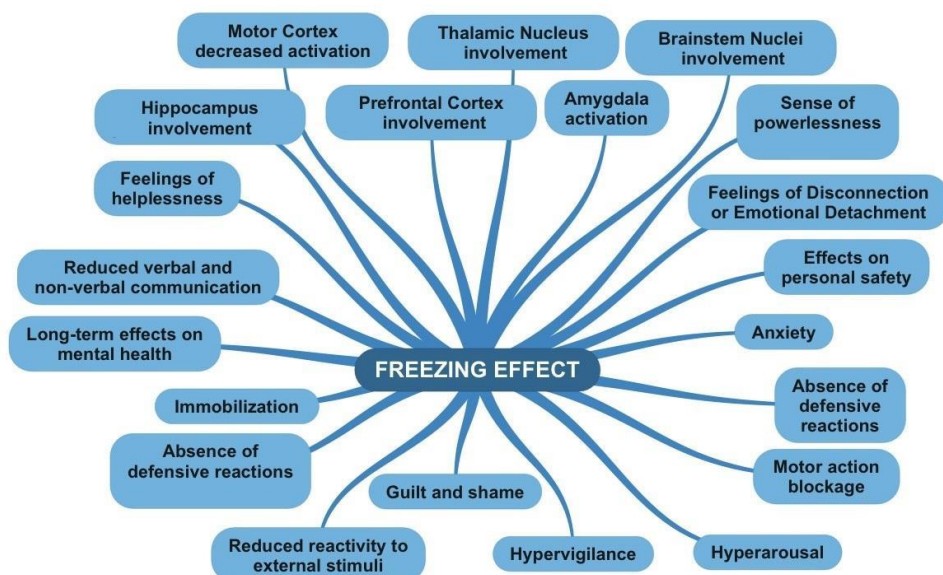

**Figure 1.** Features of freezing effect map.

## 2.1. Fight or Flight Response and Freezing Mechanism

Fight, flight, and freeze are three distinct physiological reactions that the body initiates in response to perceived danger or threat. Each of these responses is characterized by unique features. The fight response is a proactive defense mechanism activated when an individual assesses that they have sufficient resources to confront a threat. This response involves the activation of the autonomic nervous system, leading to increased muscle activity and heart rate, priming the body for potential confrontation [11].

Flight, on the other hand, is another proactive defense response. It is triggered when an individual believes they can escape the threat and reach safety. This response also activates

the autonomic nervous system, resulting in increased heart rate and lung ventilation, thus providing more oxygen to the muscles to facilitate a quicker escape [6,7].

Freeze is a passive defense response that occurs when an individual perceives no viable option for either fight or flight. This response involves the autonomic nervous system inducing a state of immobility, with reduced muscle activity, heart rate, and blood pressure. The objective is to remain inconspicuous and hope the threat passes without attack [7,8].

While fight and flight are linked to the activation of the sympathetic nervous system, which prepares the body for these active responses, they involve different neural mechanisms. The fight response is connected to the activation of the amygdala and limbic system, responsible for threat assessment and emotions like anger and aggression. It also involves the motor system and the prefrontal cortex, which play roles in movement planning and execution. The flight response, in contrast, involves the amygdala and other limbic regions like the hippocampus, crucial for memory and spatial navigation. This response also activates leg muscles and the motor area of the cerebral cortex, responsible for generating and executing movements [9,10]. In essence, while fight and flight both engage the sympathetic nervous system and certain brain regions, they represent distinct behavioral strategies in response to threats.

### 2.2. The Neural Basis of the Freezing Effect

Upon perceiving a threat, the brain activates a range of neural pathways to cope with the stressor; the autonomic nervous system (ANS) plays a key role in this process. During the freeze response, both branches of the ANS, the sympathetic and parasympathetic nervous systems, are engaged [12]. It is important to recognize that freezing's physiological characteristics are a blend of both sympathetic and parasympathetic influences, with the dominance of one system fluctuating.

The sympathetic nervous system's response is characterized by heightened alertness and physical symptoms that support freezing: increased heart rate and cardiac output, elevated blood pressure, reduced digestive activity, enhanced respiration facilitating blood flow to active tissues, muscle tension, and pain suppression [10].

Conversely, the activation of the parasympathetic nervous system during freezing often leads to slowed heart rate [13]. This dominance is marked by either a pronounced slowing of heart rate or a less significant heart rate acceleration [14].

Changes in respiratory rates and vocalizations are also linked to freezing. In rats, rapid breathing precedes ultrasonic vocalizations, which then slow down due to the longer expiration times required for these sounds. Decreased vocalizations in rats have been associated with fear in response to immediate threats, while increased vocalizations are noted in anxiety about potential threats [15].

Stress triggers the swift activation of the sympatho-adrenomedullary (SAM) system, releasing adrenaline and noradrenaline. The sympathetic nervous system's responses, including pupil dilation, accelerated heart rate, and increased muscle tone, are primarily driven by these neurotransmitters. Noradrenergic projections from the locus coeruleus to the dorsolateral periaqueductal gray (dlPAG) [16], primarily influenced by acetylcholine [17], facilitate the shift from freezing to active fear responses [18].

The hypothalamus–pituitary–adrenal (HPA) axis activation leads to the release of hormones such as corticotropin-releasing hormone (CRH), adrenocorticotropic hormone (ACTH), and cortisol (or cortisone in humans). CRH plays a crucial role in coordinating behavioral and metabolic threat responses across various brain regions, including the amygdala, and is vital for the expression of freezing in both primates and rodents [19].

Elevated levels of cortisol, whether baseline or stress-induced, have been linked to increased freezing in primates and rodents [20]. Glucocorticoids significantly contribute to developing defensive freezing. In neonatal rats, removing adrenal glands reduces freezing, which can be reversed with cortisol treatment [21]. Maternal care and postnatal adjustments in rats also decrease later cortisol responses to stress and are related to lower

freezing reactions [22]. Interestingly, a correlation has been observed in human infants between endogenous cortisol levels, freezing, and fear bradycardia, which is not seen in more sympathetically driven fear behaviors [23].

Other hormones and peptides, such as progesterone, testosterone, estrogen, oxytocin, and vasopressin, also influence freezing [24]. For example, oxytocin affects the shift from freezing to active defense by interacting with cholinergic transmission in the amygdala's lateral nucleus and the ACC, and by inhibiting vasopressin neurons in the amygdala's medial central nucleus projecting toward the ventral lateral periaqueductal gray (vlPAG) [25].

These hormones and peptides also interact with neurotransmitter systems involved in freezing, including gamma-aminobutyric acid (GABA), dopamine, and serotonin. GABA generally suppresses defensive behavior in areas like the amygdala, hypothalamus, and PAG, an effect opposed by excitatory amino acids [25]. Serotonin release in the dlPAG and ventrolateral rostral medulla inhibits active fight or flight behaviors [26].

### 2.3. Psychological Consequences of Freezing Effect: From Immediately to the Long-Term

The freezing response is associated with various anxiety and stress-related disorders, with long-term effects on cortisol levels as well [20,27]. Some studies [27,28] refer to the existence of cognitive mechanisms underlying freezing responses common in multiple forms of anxiety, which are thus considered vulnerability factors. In particular, the construct of "Looming Cognitive Style" (LCS) has been mentioned, which consists of a personal tendency to respond with freezing to threatening situations and is characterized by a predisposition to anxiety and perceiving a threat as more dangerous than it actually is. In light of this theory, individuals exhibiting "High Physical Looming" (HPL) demonstrate maladaptive behavior with a "dysfunctional freezing" response to a stimulus, regardless of the level of threat.

## 3. The Bystander Effect

### 3.1. Definition and Brief Analysis of the Phenomenon

The phenomenon of bystander inaction, commonly referred to as the bystander effect or bystander apathy, is a psychological and social occurrence where an individual observing an emergency situation fails to assist the person in distress [29]. This phenomenon is closely associated with the number of observers present; as the number of bystanders increases, the likelihood of any one individual providing help decreases. Factors contributing to the bystander effect include ambiguity, group cohesion, and a diffusion of responsibility. Darley and Latané conducted several experiments that have become keystones in social psychology. Typically, these experiments involved participants being placed either alone or amongst a group of other participants or confederates. During these sessions, an emergency situation would be simulated, and the researchers would observe and record the time taken by the participants to respond, if at all. These studies consistently demonstrated that the presence of others significantly deterred individual assistance, often by a substantial margin.

### 3.2. The Background: Kitty Genovese's Case

As is well known, psychologists John Darley and Bibb Latané [29] were the pioneers in empirically demonstrating how the presence of other people influences individual reactions in emergency situations, in a controlled laboratory environment. Their research was motivated by the 1964 case of Kitty Genovese, a New York woman who was tragically stabbed to death near her home in Queens. On the night of 13 March 1964, in New York City, at around 3:00, this 29-year-old woman was fatally stabbed near her home. She had parked her car approximately 30 m from her residence and was viciously attacked on her way home. The assailant initially fled in his car when he noticed neighbors peering out their windows, drawn in by Genovese's cries for help. However, he later returned and found Genovese in the entryway of a building, where he ultimately murdered her. The unique aspect of this crime was the number of witnesses; 38 people observed the incident from their homes, alerted by Genovese's screams. Many of these witnesses believed that

their individual intervention was unnecessary, assuming that "someone else must have seen more and already called the police," a phenomenon later termed the "diffusion of responsibility" [30].

### 3.3. From the Real World to the Laboratory

All their experiments aimed to understand the group dynamics during such incidents. In their initial experiment, college students were invited for what was presented as a casual conversation about university life and related concerns. Each participant was isolated in a room, equipped with headphones and a microphone, and communicated with others via an intercom system, a setup intended to maintain anonymity [30].

Participants were categorized into three groups based on their perceived social setting: the first group believed they were in a one-on-one conversation, the second group thought they were communicating with two others, and the third group assumed they were part of a five-person discussion. During the conversation, a participant, simulated via the intercom, appeared to suffer a convulsion and requested help. This setup allowed Latané and Darley to assess behavioral differences based on the perceived number of witnesses. Results showed that 85% of participants in the one-on-one scenario sought help, while this figure decreased to 64% when participants believed two others were present, and further dropped to 31% in the presence of four bystanders.

In a subsequent experiment, the researchers recruited students for a purported "questionnaire task". Participants were divided into two scenarios: some completed the questionnaire alone, while others did so in a room with several non-reactive collaborators (confederates). Shortly after beginning the task, black smoke began to seep out of the air conditioner, gradually filling the room. In the group with indifferent confederates, only 10% of participants left to report the smoke, taking twice as long as those who were alone, of whom 75% quickly sought help. This striking outcome corroborated the findings from the first experiment, reinforcing the idea that the likelihood of individual intervention diminishes as the number of bystanders increases.

### 3.4. Victim and Bystander: Post-Traumatic Stress Disorder

Psychological and physiological responses can arise from a variety of traumatic stressors, even those often deemed ordinary, affecting individuals who were previously in good mental health [31–36]. As is well known, repetitive abuse may affect bystanders and victims in similarly serious ways with the same levels of distress over time [33]. In fact, according to Lazarus and Folkman, psychological stress occurs when individuals perceive their interactions with their environment as potentially detrimental to their well-being. Specifically, negative evaluations of an experience, such as witnessing an act of bullying, can trigger negative emotions that lead to bystander reactions [34].

In its revised Criterion A, the DSM-5 presents a more stringent definition of trauma. This updated criterion specifies that trauma must be experienced either through direct personal involvement in the traumatic event or by being an immediate witness to such an event. It also extends to situations where individuals are informed about traumatic events that have affected close family members or friends, specifically those involving violence or accidental death. Additionally, the criterion covers repeated or intense exposure to the graphic details of severe trauma, but this exposure must be in person, thereby excluding incidents only experienced through electronic media, except in professional contexts [37]. This shift emphasizes the individual's subjective experience of trauma, suggesting that trauma is largely determined by personal perception [38]. Contemporary research is now delving into typical stressful experiences and the conditions that may lead to varying levels of trauma, acknowledging its detrimental effect on development [39].

A critical element influencing the development and severity of traumatic symptoms is repeated exposure [40–43], which greatly increases the likelihood of significant disruptions in trust and functioning. These disruptions often necessitate the re-evaluation of fundamental beliefs about oneself, others, and the world [44,45]. Such cumulative impacts can affect

every aspect of human functioning, including physical and emotional health, personality, social relationships, cognitive abilities, spirituality, and behavior [46–48].

Children and adolescents exhibit a heightened vulnerability to traumatic experiences, which can exert a significant and enduring impact on their developmental trajectory [49–51]. Research suggests that exposure to stressors, which may be perceived as normal or minor, such as bullying, harassment, or teasing, can precipitate long-term detrimental consequences [52–56]. Chronic exposure in this demographic is associated with amplified distress and an increased manifestation of symptoms [57–61]. These experiences hold potential implications for behavioral patterns and neurobiological development [57]. Despite these risks, adults often downplay these experiences, viewing them as minor forms of trauma, even considering them part of normal development [58]. This attitude results in limited support for direct victims and none for bystanders, leading children and adolescents to feel that adults are indifferent, unaware, or even approving of such abuse [59,60].

In the 1990s, the connection between such forms of abuse and youth suicides [61] and school violence [62] gained increased recognition. Childhood experiences of low-level abuse are no longer seen as trivial or harmless but are acknowledged for their potential harm to both the abused and those around them. The impact extends beyond direct victims to include bystanders, who can experience traumatic responses similar to those of the victims, blurring the line between victim and observer [63]. Studies have noted similarities in symptoms between bystanders and victims, including physiological arousal, reduced empathy [64], desensitization to negative behaviors in schools [65–67], and overall risky and negative behaviors [53,55]. Common feelings of isolation and inefficacy [53] are also observed. There is a growing argument for considering witnesses as co-victims or indirect victims [66], highlighting the need to recognize and support those indirectly affected by traumatic events (see Figure 2).

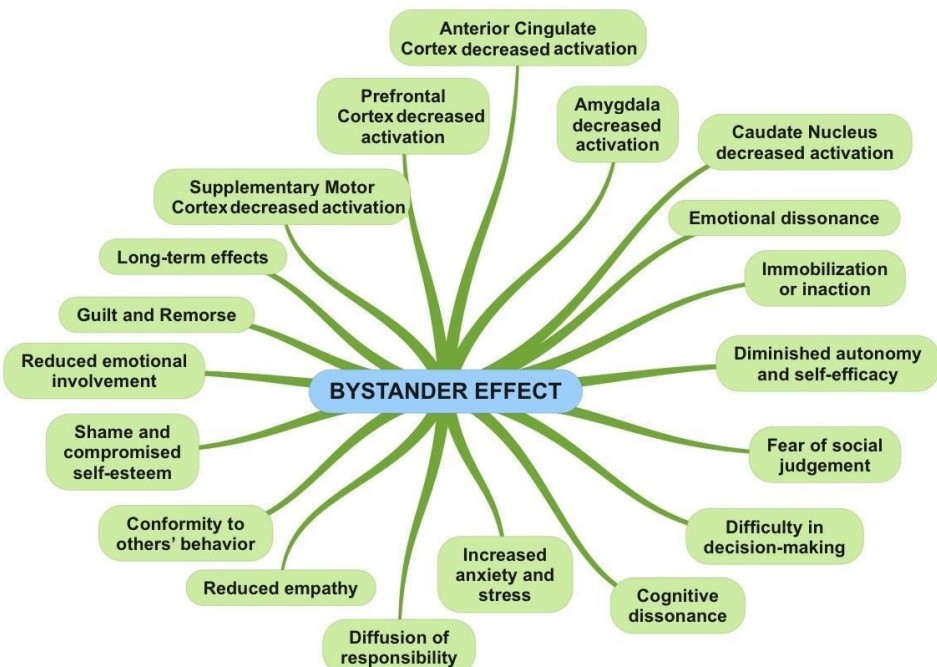

**Figure 2.** Features of bystander effect map.

### 3.5. Psychological Consequences in Bystander Effect: Now and in the Future

According to a study by Itzkovich et al., bystanders in cases of bullying are indirect victims, "by proxy", a position that affects their well-being and psychological health. In fact, a correlation has been demonstrated between witnessing an act of bullying and suicidal ideation, depressive symptoms, anxiety, and stress, as well as repression of empathy among bystanders and increased feelings of guilt [65]. The theoretical model employed by the

authors is the COR (conservation of resources) theory, through which it is possible to observe bystanders' reactions to bullying in light of their individual coping resources, which help reduce exposure to stressors. Knauf et al. [66] focused on various determinants such as moral disengagement, empathy, self-efficacy, and feelings of responsibility as underlying cognitive–affective processes preceding bystanders' reactions. Bandura [67] argued about moral disengagement as a mediator between moral reasoning and action, and as a self-regulation process to decrease tension associated with adopting behaviors contrary to morality/ethics. Byers [68] suggests that bystanders tend to resort to moral disengagement due to feelings of anxiety and frustration as coping mechanisms. Self-efficacy is supposed to influence bystander behavior according to various research; in particular, people with a lack of confidence exhibit this behavior more frequently instead of protective/proactive behaviors [69]. On the other hand, a high level of social support and of personal self-efficacy promotes defender behavior instead of bystander behavior.

## 4. A Comparison between the Two Phenomena

As seen in the previous paragraphs, the bystander effect and freezing phenomenon have various points of contact, such as behavior, but also some necessary distinctions, for example, in the neural regions involved or in terms of affective reactions. From a theoretical perspective, despite being two mechanisms of great interest in the field of social psychology, there are no studies that have attempted to systematically compare these two constructs. The purpose of this study was to compare these two mechanisms, starting from the definition, with respect to emotional, behavioral, neurobiological reactions, and psychological outcomes. The following table (Table 1) represents the details of the comparison for each sub-category and highlights points of contact and differences.

**Table 1.** Comparison between the two Phenomena.

| Bystander Effect | Freezing Effect |
|---|---|
| **Definition:**<br>It occurs in emergency or danger situations involving another person, resulting in a lack of action. | **Definition:**<br>It occurs in emergency or danger situations involving the person, resulting in a lack of action. |
| **What happens on an emotional level?**<br><br>**Fear of social judgment:** Individuals may experience fear of being judged or evaluated by others present. They might worry about appearing foolish, intrusive, or doing something wrong. This fear of social judgment can hinder the expression of emotions and action, leading to feelings of anxiety or shame [63].<br><br>**Emotional dissonance:** In some cases, people may experience emotional dissonance when their personal emotional reaction conflicts with that of others present. For instance, an individual might feel concerned or compassionate for someone in distress, but if others appear indifferent or do not react, emotional tension may arise. This disparity between personal emotions and those of others can generate emotional discomfort or frustration. | **What happens on an emotional level?**<br>**Hypersensitivity and Hypervigilance:** In some situations, freezing may be accompanied by an increased sensitivity to stimuli in the surrounding environment. People can be hyperaware and hypersensitive to danger signals or any changes in the situation, attempting to detect any potential threat [64].<br><br>**Sense of powerlessness:** During freezing, individuals may feel powerless or unable to act. This can generate frustration, resignation, or a sense of being trapped in the situation with no way out [64].<br><br>**Feelings of Disconnection or Emotional Detachment:** During freezing, some individuals may experience a sense of emotional disconnection or detachment from the situation. This can be a form of psychological defense that allows them to cope with danger or threat without being overwhelmed by the intense emotions associated with them. |
| **What happens at the behavioral level?**<br>**Immobilization or inaction:** The bystander effect manifests through the immobility or inaction of individuals involved. People may remain passive and refrain from taking any action to help or intervene in the situation of danger or emergency. | **What happens at the behavioral level?**<br>**Immobilization:** During freezing, the individual may remain still and frozen in the position they were in at the moment of perceiving danger. The lack of voluntary movements is a key characteristic of freezing [66]. |

**Table 1.** *Cont.*

| Bystander Effect | Freezing Effect |
|---|---|
| **Difficulty in decision-making:** People in the bystander effect may experience difficulty in making decisions regarding the action to take. They may feel indecisive about what to do or may seek guidance or initiative from others [65]. | **Absence of defensive reactions:** Unlike other defense responses such as flight or fight, in freezing, the individual shows no active reaction to protect themselves or avoid the danger. There is no attempt to escape the threat or defend against it. |
| **Cognitive dissonance:** The bystander effect can generate cognitive dissonance, a discrepancy between what a person knows is right (helping someone in danger) and their actual behavior (remaining still or inactive). This discrepancy can create a sense of emotional discomfort and ambivalence. | **Reduced verbal and non-verbal communication:** During freezing, the individual may exhibit reduced verbal and non-verbal communication. Gestures, facial expressions, or words may be limited or absent as energy and attention are focused on maintaining immobility [1]. |
| **Diffusion of responsibility:** Another behavioral aspect of the bystander effect is the diffusion of responsibility. People tend to feel less responsible to intervene if they are surrounded by others, if someone else will take care of the situation. | **Motor action blockage:** Freezing is characterized by a blockage of motor actions. The individual may temporarily lose the ability to move or perform tasks that require voluntary action. |
| **Conformity to others' behavior:** The bystander effect can lead to conformity to the behavior of others present in the situation. People may observe the behavior of others and model their reaction based on what others are doing or not doing. | **Reduced reactivity to external stimuli:** During freezing, the individual may show reduced reactivity to external stimuli. They may be less sensitive to sounds, voices, or surrounding events as attention is focused on the perceived danger or threat. |
| **Reduced emotional involvement:** People in the bystander effect may experience reduced emotional involvement in the danger or emergency. Since there are other people present who are not reacting, the individual may feel less emotionally engaged or less motivated to intervene. | **Increased hypervigilance:** Despite immobility, the individual may exhibit increased hypervigilance toward the surrounding environment. They may be hypersensitive to danger signals and maintain a high state of alertness for potential threats. |
| *Which brain areas are activated?* | *Which brain areas are activated?* |
| **Prefrontal Cortex:** The prefrontal cortex is involved in planning, processing social information, and assessing risks. Under the bystander effect, reduced activation of the prefrontal cortex has been observed, which could be correlated with a decrease in individual motivation or attention toward the situation [67]. | **Prefrontal Cortex:** The prefrontal cortex is involved in many higher cognitive functions, including evaluation, planning, and emotional control. During freezing, the prefrontal cortex may be engaged in evaluating the situation and regulating emotional responses [10]. |
| **Amygdala:** The amygdala is involved in the emotional response, particularly in the detection and processing of emotions such as fear. During the bystander effect, the amygdala may show reduced activation, as individual emotional engagement may be attenuated by the presence of other people. | **Amygdala:** The amygdala is a key region involved in the fear and threat response. During freezing, the amygdala may show increased activation in response to the perception of danger or threat. |
| **Anterior Cingulate Cortex:** The anterior cingulate cortex is involved in emotion regulation, attention, and the evaluation of error or conflict situations. Under the bystander effect, the activation of the anterior cingulate cortex may be reduced, suggesting decreased awareness, or monitoring of emergency situations. | **Hippocampus:** The hippocampus is involved in memory and learning. During freezing, the hippocampus may be involved in processing and remembering information related to the dangerous or threatening situation. |
| **Caudate Nucleus:** The caudate nucleus is involved in the decision-making process and regulation of behavior. Under the bystander effect, decreased activation of the caudate nucleus has been observed, which may be related to reduced motivation for action or inhibition of behavioral responses. | **Thalamic Nucleus:** The thalamic nucleus plays a role in transmitting sensory information and regulating attention. During freezing, the thalamic nucleus may be involved in filtering and transmitting sensory information relevant to the perception of danger. |
| **Supplementary Motor Cortex:** The supplementary motor cortex is involved in the planning and execution of voluntary movements. Under the bystander effect, the supplementary motor cortex may show reduced activation, as immobility and the inhibition of motor responses are characteristic of the bystander effect. | **Brainstem Nuclei:** Brainstem nuclei, such as the locus coeruleus and the raphe nucleus, are involved in regulating arousal and physiological responses to stress. During freezing, these nuclei may be activated to prepare the body to respond to the threat or danger. |
| | **Motor Cortex:** The motor cortex is involved in the generation and execution of voluntary movements. During freezing, the motor cortex may show reduced activation, as freezing involves immobility and the inhibition of motor responses. |

**Table 1.** *Cont.*

| Bystander Effect | Freezing Effect |
|---|---|
| *Psychological Consequences:* | *Psychological Consequences:* |
| **Increased anxiety and stress:** The bystander effect can lead to increased emotional anxiety and stress. Awareness of the danger or the responsibility to intervene can cause a sense of agitation and worry [65]. | **Anxiety and hypervigilance:** After freezing, some individuals may develop increased anxiety and hypervigilance. They may become hypersensitive to danger signals, overly alert, and constantly vigilant of potential threats [20,28]. |
| **Reduced empathy:** The bystander effect may result in reduced empathy toward the victim or the person in danger. The presence of others who do not react can create a social climate where individual empathy is suppressed or minimized [65,66]. | **Feelings of helplessness:** During freezing, individuals may experience a sense of helplessness or an inability to act. This can lead to frustration and a loss of confidence in one's ability to handle dangerous situations. |
| **Diminished autonomy and self-efficacy:** Being part of the bystander effect can undermine the sense of autonomy and control over one's life. People may feel powerless or unable to make decisions and act independently, creating a perception of reduced self-efficacy [70]. | **Effects on personal safety:** Freezing can have consequences for the perception of personal safety. After experiencing freezing, people may feel more vulnerable or insecure about their ability to defend themselves or handle similar situations in the future. |
| **Guilt and Remorse:** Individuals experiencing the bystander effect may experience a profound sense of guilt and remorse for not taking action or providing help when necessary. These feelings may come from realizing that not acting could have made the situation worse or caused harm to the victim [65,66]. | **Guilt and shame:** After freezing, some individuals may experience guilt or shame for not reacting or taking action to protect themselves or others. These feelings may stem from the perception of having failed to address the situation or fulfill their duty. |
| **Shame and compromised self-esteem:** Being a spectator in a situation where someone is in danger can generate shame and a sense of compromised self-esteem. Individuals may feel inadequate or incapable of intervening, negatively impacting their self-perception [69,70]. | **Long-term effects on mental health:** In some cases, the experience of freezing can have long-term consequences on mental health. Feelings of helplessness, guilt, or shame can contribute to the development of psychological disorders such as anxiety, depression, or post-traumatic stress disorder (PTSD) [31,33]. |
| **Long-term effects:** In some cases, the experience of being involved in the bystander effect can have long-term consequences on mental health. Persistent guilt, shame, and remorse can contribute to the development of disorders such as anxiety, depression, or post-traumatic stress disorder [65,66,69,70]. | |

## 5. Discussion

The results of this comparison indicate that the bystander effect and freezing share numerous points of contact but also have some distinctive mechanisms.

From an emotional standpoint, the bystander effect is characterized by the fear of social judgment and emotional dissonance, while freezing is characterized by a sense of helplessness, hypersensitivity, hypervigilance, and feelings of emotional detachment [71–74].

From a behavioral perspective, both the bystander effect and freezing involve immobility as a behavioral response, but they differ regarding decision-making difficulty, cognitive dissonance, the role of the diffusion of responsibility, conformity to others' behavior, and emotional involvement [75–77].

In terms of neurobiology, there are some similarities and differences between the bystander effect and freezing in the brain areas involved. Both effects show reduced activation of the prefrontal cortex, which is involved in planning, processing social information, and risk assessment. In the bystander effect, the reduced activation might be related to decreased individual motivation or attention to the situation. In freezing, on the other hand, the prefrontal cortex might be involved in evaluating the situation and regulating emotional responses [78].

Another area that shows similarities is the activation of the amygdala, involved in emotional response and the detection of emotions such as fear. However, in the bystander effect, reduced amygdala activation is observed, likely due to the attenuation of individual emotional involvement determined by the presence of other people. In contrast, in freezing, the amygdala may show increased activation in response to the perception of danger or threat [79,80].

The motor cortex is another area where similarities are observed. In both the bystander effect and freezing, reduced activation of the motor cortex is observed. This is due to the common characteristics of immobility and the inhibition of motor responses in both phenomena.

Differences emerge in other areas involved. In the case of the bystander effect, reduced activation of the anterior cingulate cortex, which is involved in regulating emotions, attention, and evaluating situations of error or conflict, is observed. This reduced activation suggests lower awareness or monitoring of emergency situations in the bystander effect. In freezing, however, the hippocampus is involved in processing and remembering information related to the situation of danger or threat [75].

Other differences are found in the involvement of the thalamus nucleus and brainstem nuclei. In freezing, the thalamus nucleus plays a role in filtering and transmitting sensory information relevant to the perception of danger, while brainstem nuclei, such as the locus coeruleus and the raphe nucleus, are activated to prepare the body to respond to the threat or danger. In the case of the bystander effect, however, the activation of the thalamus nucleus and brainstem nuclei is not specifically mentioned.

In summary, the bystander effect and freezing show similarities in the brain areas involved, such as the prefrontal cortex, amygdala, and motor cortex. However, they differ in other involved regions, such as the anterior cingulate cortex, hippocampus, thalamus nucleus, and brainstem nuclei, which show specific activations for each phenomenon.

Finally, the psychological consequences of the bystander effect and freezing also have some similarities and differences. Both effects, the bystander effect and freezing, can generate a sense of guilt and remorse. Those involved may feel responsible for not taking action or providing help when necessary, leading to feelings of guilt and remorse [81]. In both cases, these feelings can stem from an awareness of their own inaction and the potential contribution to the worsening of the situation or harm to the victim. Shame and a compromised sense of self-esteem are common psychological consequences in both the bystander effect and freezing. Those involved may feel inadequate or unable to intervene, negatively impacting their self-perception. This sense of shame and compromised self-esteem can be fueled by individuals' awareness of themselves being spectators in a dangerous situation or not reacting appropriately.

The increase in anxiety and stress is another shared consequence in both the bystander effect and freezing. Awareness of the dangerous situation or the responsibility to intervene can lead to a sense of restlessness and worry. In both cases, those involved may experience heightened anxiety and emotional stress related to the situation.

However, there are also some specific psychological consequences for each phenomenon. In the case of the bystander effect, there is a reduction in empathy toward the victim or the person in danger. The presence of other people who do not react can create a social climate where individual empathy is suppressed or minimized [71]. In freezing, on the other hand, there is a sense of helplessness or inability to act, which can lead to frustration and a loss of confidence in one's ability to handle dangerous situations. Furthermore, in the bystander effect, there is a reduction in autonomy and self-efficacy. Those involved may feel powerless or incapable of making decisions and acting independently, creating a perception of reduced self-efficacy. In freezing, effects on personal safety can develop, with a perception of vulnerability or insecurity about one's ability to defend oneself or face similar situations in the future.

Finally, both effects can have long-term consequences on mental health. In the case of the bystander effect, persistent guilt, shame, and remorse can contribute to the development of disorders such as anxiety, depression, or post-traumatic stress disorder. In freezing, feelings of helplessness, guilt, or shame can have similar long-term effects on mental health, contributing to the development of disorders like anxiety, depression, or post-traumatic stress disorder [82–86].

## 6. Limitations

This comparative study between the freezing effect and the bystander effect presents several significant limitations that influence its scope and the interpretation of results. First and foremost, being of a psychological nature, this study does not incorporate genetic or evolutionary aspects that could provide additional explanations for the differences observed between the two phenomena. This approach omits a potential dimension of analysis, limiting understanding to a purely behavioral and psychological context without considering the influence of biological or historical factors. Secondly, the investigation of the freezing and bystander effects encounters significant ethical challenges when attempting to study them in real-world contexts. As a result, this study is primarily based on a theoretical synthesis rather than on solid empirical data. This lack of direct experimentation and observation in real-life situations limits the ability to draw definitive conclusions, suggesting the need for further research that explores alternative methods to study these behaviors in a more thorough and ethically responsible manner.

## 7. Conclusions

Today, news reports often cover cases of bullying, cyberbullying, and other dramatic episodes where, despite the presence of witnesses, intervention is not always timely. In some cases, there is proactive intervention, while in others, the bystander does not intervene, either due to the bystander effect or the freezing effect; therefore, the same action can be interpreted in two different ways. The study allows for a better understanding of the underlying implications and overlapping aspects of how both phenomena can lead to avoidance behaviors in emergency situations. Therefore, it proves to be enlightening across multiple domains compared to other studies that do not make comparisons between the two phenomena and that address them separately.

## 8. Notes on Preventive Strategies

The prevention of the bystander effect can be addressed through a series of strategies aimed at promoting a culture of responsibility and solidarity. These measures are crucial to counteracting the inertia and immobility that often characterize situations where many people are present, but none feel personally involved in providing help [80,84,86].

A first step in preventing the bystander effect is awareness and information. It is essential to raise awareness about the bystander effect, explaining its mechanisms and the negative consequences it can entail. Awareness campaigns, educational programs in schools and workplaces, and the dissemination of information through the media can contribute to making people aware of the bystander effect and the importance of taking action in emergency situations [87–89].

Promoting individual responsibility is another key aspect in preventing the bystander effect. People should be encouraged to consider themselves responsible for intervening or providing help when facing dangerous situations. This can be achieved by promoting positive messages about the importance of individual action and the power that each person has to make a difference.

Another effective approach is empathy and leadership training. Providing training opportunities that develop empathy and leadership skills can help individuals overcome the bystander effect. These programs can enhance awareness of others' emotions and promote the courage to act in challenging situations [89]. Additionally, encouraging people to develop leadership skills can foster initiative and a sense of responsibility. Creating a culture of support and collaboration is a crucial element in preventing the bystander effect. When people feel part of a community where mutual support and collaboration are fundamental values, the conditions for more proactive action in emergency situations are established. Encouraging mutual aid and promoting a culture of solidarity can motivate people to intervene and provide support when needed [90].

Preventing freezing, which is the paralysis or inability to act in dangerous situations, requires the adoption of various strategies. Initially, it is crucial to raise awareness and

readiness among individuals. This involves acquiring knowledge about the phenomenon of freezing, understanding its potential negative consequences, and becoming familiar with the signs indicating its presence. Being informed and mentally prepared helps people to recognize freezing when it occurs and respond appropriately [89].

Another effective strategy is training and simulations. Organizing exercises and practical simulations allows individuals to experience controlled dangerous situations and develop the necessary skills to face them. During these training sessions, it is possible to become acquainted with the dynamics of freezing and learn hazard management strategies [27].

Stress management is another crucial aspect in preventing freezing. Stress and anxiety can inhibit he ability to act in critical situations. Therefore, learning stress management techniques, such as deep breathing, meditation, or physical exercise, can help maintain composure and reduce the emotional impact of dangerous situations. Effective stress management promotes a greater capacity to react and make decisions quickly and efficiently [91].

Enhancing self-efficacy and self-confidence is another important strategy for preventing freezing. Developing a sense of confidence in one's ability to handle challenging situations is essential. This can be achieved through accomplishing personal goals, acquiring specific skills, and gradually facing situations that pose a challenge. When one has self-confidence, they are more ready to take action and reduce the inability to cope with dangerous situations.

**Author Contributions:** Conceptualization, A.R. and E.S.; methodology, A.R.; software, E.S.; validation, G.I., M.B. and A.M.; formal analysis, E.S.; investigation, A.R., E.S., G.I. and M.B.; resources, A.M.; data curation, E.S.; writing—original draft preparation, A.R.; writing—review and editing, M.B.; visualization, F.B. and G.T.; supervision, A.R. All authors have read and agreed to the published version of the manuscript.

**Funding:** This research received no external funding.

**Institutional Review Board Statement:** Not applicable.

**Informed Consent Statement:** Not applicable.

**Data Availability Statement:** Not applicable.

**Conflicts of Interest:** The authors declare no conflicts of interest.

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
