# Peer review of "Freezing Effect and Bystander Effect: Overlaps and Differences"

_psych, doi:10.3390/psych6010017_

Round 1

Reviewer 1 Report

Comments and Suggestions for Authors

I have not comments and suggestions to the Authors!

Author Response

Dear Reviewer,

thank you for your positive evaluation and to take time to review.

Reviewer 2 Report

Comments and Suggestions for Authors

The manuscript "Freezing Effect and Bystander Effect: Overlaps and Differences" is an interesting and valuable work comparing two similar phenomena. I appreciate the work put into preparing the manuscript and the detailed analysis of both psychological phenomena, regarding cognitive, affective and behavioral aspects. The table is particularly useful. However, the work requires improvement.

1. Some parts of the manuscript are missing, such as authors' names, or a section describing limitations of the study.

2. The other problems are that the authors used in title "Section 1", "Section 2" and so on, which is redundant, since hierarchical headings are used here. 

3. Many times names and forenames of some researchers are used without any references (e.g., lines: 137, 156, 160, 173).

4. Reference style is inconsistent, especially in Table 1 both APA and MDPi styles are used (e.g., "Gottlieb et al. (1976)" as APA  and "[62]" as MDPI). Please correct it.

5. My main substantive concern is that biological and medical approaches are overrepresented in the description of the freezing effect, while psychological theories are missing. Gray's theory is nowhere highlighted and described in detail, although this is the most important approach to the freezing effect in psychology of individual differences. Please add such a description so that the reader can better understand the issues in relation to people (instead of animals, as current references 1-7 suggest in this work).

Author Response

The manuscript "Freezing Effect and Bystander Effect: Overlaps and Differences" is an interesting and valuable work comparing two similar phenomena. I appreciate the work put into preparing the manuscript and the detailed analysis of both psychological phenomena, regarding cognitive, affective and behavioral aspects. The table is particularly useful. However, the work requires improvement.

Thank you for your evaluation and time

Some parts of the manuscript are missing, such as authors' names, or a section describing limitations of the study.

As regards authors’ names I’m not sure you received a blind review. However we repeated the same affiliation for all authors, by adding personal e-mail.

The study limitation section was added, thank you for noticing.

  1. The other problems are that the authors used in title "Section 1", "Section 2" and so on, which is redundant, since hierarchical headings are used here. 

True. We agree with you, we removed "Section 1", "Section 2" and so on, leaving only principal headings.

  1. Many times names and forenames of some researchers are used without any references (e.g., lines: 137, 156, 160, 173).

We completely revised the half of reference list. We carefully verify each reference number and citing appropriately in the text.

  1. Reference style is inconsistent, especially in Table 1 both APA and MDPi styles are used (e.g., "Gottlieb et al. (1976)" as APA  and "[62]" as MDPI). Please correct it.

We changed the table by choosing MDPI style.

  1. My main substantive concern is that biological and medical approaches are overrepresented in the description of the freezing effect, while psychological theories are missing. Gray's theory is nowhere highlighted and described in detail, although this is the most important approach to the freezing effect in psychology of individual differences. Please add such a description so that the reader can better understand the issues in relation to people (instead of animals, as current references 1-7 suggest in this work).

When changing tables, we also added Gray’s theory as opening paragraph, with the reference to freezing theory. We believe that moving this information in the main text could be the best choice to organize informations, thank you for your suggestion, it was really helpful and appreciated.

Thank you for your help!

Dr. Amelia Rizzo,

The corresponding author

Reviewer 3 Report

Comments and Suggestions for Authors

This study used ChatGPT for generating ideas and cannot be published.

Author Response

Dear reviewer,

thank you for arising this important question.

The paper was originally written in italian,

then translated in English with chat GPT.

This is the reason why I guess you didn't find grammatical mistakes.

However the research idea and the comparison is a product of our works as researcher.

Maybe it can be found some trace of translation activity,

but when checked with AI detector, the paper result 100% original.

You can find in attachment the AI detector report.

Kind regards,

Dr. Amelia Rizzo

Round 2

Reviewer 2 Report

Please add the "Conclusions" section, summarizing the importance of the study and literature gap, and explaining main findings, its interpretation, and practical implications, as a short message to the reader.

References to Gray's theory are still missing. Similarly, many references are missing to psychological explanations of stress, PTSD, emotions, cognitive and behavioral responses to threat. The neurobiological approach is overused, but the psychological approach is lacking. The table is inadequate to the content in the introduction. The authors should develop all relevant information contained in the Table in several sections and subchapters (including the neurobiological or psychophysiological approach, and separately the psychological approach, including in-depth cognitive, emotional and behavioral theories), based on the psychological literature to each definition and other dimensions. The table should only be a summary and not contain a lot of new information, not supported by evidence in the form of research and theory, within references to the literature.

The manuscript requires restructuration and better literature review.

Author Response

In this revised file, references for each concept illustrated in the table regarding the psychological consequences of both phenomena have been included. Two brief sections on this topic have been added, redundant parts regarding the Kitty Genovese incident have been removed, conclusions have been inserted, bibliographic references in the text have been reorganized, and further elaboration has been provided where necessary (such as regarding Gray's theory). Additionally, a brief connection to PTSD has been included. The changes are highlighted in cyan.

Thank you for your support!

Reviewer 3 Report

It should be emphasised that prestigious journals generally do not accept articles translated solely by AI. Researchers are advised to personally supervise the translation and employ the services of a proofreader to ensure accuracy and linguistic improvement. This practice ensures that the final version of the article will meet the high standards and expectations set by reputable journals.

It should be emphasised that prestigious journals generally do not accept articles translated solely by AI. Researchers are advised to personally supervise the translation and employ the services of a proofreader to ensure accuracy and linguistic improvement. This practice ensures that the final version of the article will meet the high standards and expectations set by reputable journals.

Author Response

In this revised file, references for each concept illustrated in the table regarding the psychological consequences of both phenomena have been included. Two brief sections on this topic have been added, redundant parts regarding the Kitty Genovese incident have been removed, conclusions have been inserted, bibliographic references in the text have been reorganized, and further elaboration has been provided where necessary (such as regarding Gray's theory). Additionally, a brief connection to PTSD has been included. The changes are highlighted in cyan.

Thank you for your support and Ethical issue arised. We agree with you.

Round 3

Reviewer 2 Report

The authors significantly improved the manuscript.

Thank you for your cooperation in improving the manuscript.

Reviewer 3 Report

The revised version is fine in terms of content.

The revised version is fine in terms of content.